# Cystic Fibrosis in Adults: A Paradigm of Frailty Syndrome? An Observational Study

**DOI:** 10.3390/jcm13020585

**Published:** 2024-01-19

**Authors:** Paola Iacotucci, Vincenzo Carnovale, Lorenza Ferrillo, Jolanda Somma, Marialuisa Bocchino, Marcella D’Ippolito, Alessandro Sanduzzi Zamparelli, Giuseppe Rengo, Nicola Ferrara, Valeria Conti, Graziamaria Corbi

**Affiliations:** 1Department of Clinical Medicine and Surgery, University of Naples “Federico II”, 80138 Naples, Italy; paola.iacotucci@unina.it (P.I.); marialuisa.bocchino@unina.it (M.B.); sanduzzi@unina.it (A.S.Z.); 2Department of Translational Medical Sciences, University of Naples “Federico II”, 80138 Naples, Italy; lorenza.ferrillo@unina.it (L.F.); jolanda.somma@unina.it (J.S.); marcella.dippolito@unina.it (M.D.); giuseppe.rengo@unina.it (G.R.); nicferra@unina.it (N.F.); graziamaria.corbi@unina.it (G.C.); 3Department of Medicine Surgery and Dentistry, Scuola Medica Salernitana, University of Salerno, Baronissi, 84084 Salerno, Italy; vconti@unisa.it

**Keywords:** functional status, frailty, Cystic Fibrosis, geriatric syndrome

## Abstract

This study aimed to assess the main clinical and anamnestic characteristics of adult Cystic Fibrosis (CF) patients and to evaluate the association of frailty with the CF genotyping classification. In an observational cross-sectional study, all ambulatory CF patients over 18 years old who received a diagnosis at the Regional Cystic Fibrosis Center for adults were enrolled and assessed by spirometry for respiratory function, by ADL and IADL for functional status, and by the Study of Osteoporotic Fractures (SOF) Index for frailty. The study population consisted of 139 CF patients (mean age 32.89 ± 10.94 years old, 46% women). Most of the subjects were robust (60.4%). The pre-frail/frail group was more frequently females (*p* = 0.020), had a lower BMI (*p* = 0.001), worse respiratory function, a higher number of pulmonary exacerbations/years, cycles of antibiotic therapy, and hospitalization (all *p* < 0.001) with respect to robust patients. The pre-frail/frail subjects used more drugs and were affected by more CF-related diseases (all *p* < 0.001). In relation to logistic regression, the best predictor of the pre-frail/frail status was a low FEV1 level. The CF patients show similarities to older pre-frail/frail subjects, suggesting that CF might be considered an early expression of this geriatric syndrome. This finding could help to better define the possible progression of CF, but overall, it could also suggest the usefulness employing of some tools used in the management and therapy of frailty subjects to identify the more severe CF subjects.

## 1. Introduction

Cystic Fibrosis (CF) is caused by mutations in the CF Transmembrane Conductance Receptor (CFTR) gene [1] that translates into a defective protein with the loss of activity [1,2]. Due to its multi-systemic distribution, the disease affects organs and tissues where CFTR is expressed. CF prevalence varies by country, with the Europe accounting for 44,719 patients and Italy for 5801 of all affected individuals [3]. Because it is an autosomal recessive inherited disorder, for many decades, CF was considered a childhood disease, but in recent years, multifactorial improvements have increased the survival of CF patients. In fact, by 1990, the median survival had increased to 28 years, and it currently exceeds 40 years [4]. Consequently, in 2021, the child/adult ratio affected by this condition in Italy was 48 vs. 52% [5].

Otherwise progress in diagnostic techniques, with the inclusion of genetic tests, has also led to the identification of other atypical clinical CF forms, which are frequently monosymptomatic and are mainly diagnosed in adulthood with frequent negative or dubious results on sweat tests [6,7]. Following the latest guidelines [8], a CF diagnosis can be considered when an individual has both a clinical presentation of the disease and evidence of CFTR dysfunction. The tests of CFTR function are not always performed in the following order, but are still performed hierarchically to establish the diagnosis of CF: sweat chloride should be considered first, then CFTR genetic analysis, and then CFTR physiologic tests. All individuals diagnosed with CF should have a sweat test and a CFTR genetic analysis performed. Rare individuals with a sweat chloride of <30 mmol/L may be considered to have CF if alternatives are excluded, and the other confirmatory tests (genetic and physiologic testing) support a CF diagnosis.

If only one CFTR variant is identified based on limited analyses, further (“extended”) CFTR testing should be performed. CF is possible if both alleles possess CF-causing, undefined, or mutation of varying clinical consequence (MVCC) mutations; however, CF is unlikely if only non-CF-causing mutations are found. If a CF diagnosis is not resolved, Cystic Fibrosis Screen Positive, Inconclusive Diagnosis/CFTR-related metabolic syndrome (CFSPID/CRMS) (following newborn screening or CFTR-related disorder) should be considered. Rarely, no distinct label may be appropriate, but further follow-up may be warranted. In these cases, the “CF carrier” or the specific clinical problem should be used for characterization/labeling purposes [8].

When applied to adults, all patients must have one or more CF symptoms combined with evidence of CFTR dysfunction, determined through sweat testing and/or genotype analysis. Despite its apparent simplicity, applying the criteria is sometimes difficult in the clinical care setting [8].

Frailty represents a geriatric syndrome generally recognized as a state of high vulnerability due to adverse health outcomes, including an increased risk of falls and fractures, hospitalization, decreased quality of life, physical disability, iatrogenic complications, and early mortality [9,10,11]. Indeed, frailty still represents a condition conceptually confined to the geriatric population, whereas some CF patients seem to be eligible to be defined as frail. 

Common findings such as an inflammatory status and sarcopenia could suggest a common substrate in both conditions. In a systematic review, Calella et al. [12] underlined that patients with CF are affected by respiratory and gastrointestinal problems including airway inflammation and an increased susceptibility to repeated infections, exocrine pancreatic insufficiency and malabsorption, and a chronic inflammatory status that leads to general discomfort and a poor quality of life [12]. Moreover, reduced physical activity and increased resting energy expenditure could represent other factors implicated in the genesis of sarcopenia in CF patients. 

All the aforementioned factors are similarly called into question when determining sarcopenia. Inflammation may accelerate the catabolism of skeletal muscles and adipose tissue, inducing the muscle weakness and weight loss that are characteristic of frailty [13].

Until now, only a few small studies have examined the characteristics of adult CF patients and suggested possible similarities in the frailty syndrome of specific elderly CF patients. This study aimed to assess the main clinical and anamnestic characteristics of an adult Cystic Fibrosis population and to evaluate the possible association of frailty syndrome with these characteristics and the genotyping classification.

## 2. Materials and Methods

### 2.1. Study Population

This was an observational cross-sectional study on 139 ambulatory CF patients who received a diagnosis at the Regional Cystic Fibrosis Center for Adults (Department of Translational Medical Sciences, University of Naples Federico II) and were assessed by the center from January 2022 to May 2023. All patients who met the diagnostic criteria for CF [8], namely over 18 years old, with specific pathological sweat chloride levels (chloride > 60 mEq/L) and two CFTR mutations, were recruited. Key exclusion criteria included mechanical ventilation, severe liver disease with or without hepatic impairment, a history of solid organ or hematological transplantation, a history of drug or alcohol abuse in the past year, and pregnancy.

Sweat chloride levels were tested as previously described [14], and a panel of CFTR mutations was screened. The CFTR genotype was defined through the screening of the most frequent mutations and rearrangements. Specifically, it was defined by a first-level molecular analysis [14], and in all cases, gene sequencing excluded other CIS mutations [15,16].

All participants were assessed by spirometry for the following parameters: Forced Expiratory Volume in 1 s (FEV1—L); Forced Expiratory Volume in 1 s percent predicted (ppFEV1%); Forced Vital Capacity (FVC—L); Forced Vital Capacity percent predicted (FVC %); and Maximal Mid-Expiratory Flow (MMEF). 

The Ethical Committee of the University Federico II of Naples approved the protocol (Ref. 27/202), and the study was registered with the ClinicalTrials.gov Registry (NCT06020547). Informed consent from each recruited subject was obtained. The research protocol was conducted following the principles of the Helsinki II Declaration and its amendments. This report was written according to the STROBE guidelines for Observational Studies in Epidemiology [17] (see Appendix A).

### 2.2. Functional Status and Anthropometric Measurements

Functional status was assessed by evaluating activities of daily living (ADLs) [18], and Instrumental ADLs (IADLs) [19]. Dependence on one or more IADLs indicates that a person cannot safely live alone; dependence on one or more ADLs indicates that a person needs assistance for some specific activities that are necessary for survival. The participants’ height and weight were also recorded, and their Body Mass Index (BMI) was calculated as body weight divided by height squared (kg/m^2^). Body weight was measured in a fasting state in the morning with a mechanical column scales with eye-level beam balance (±0.1 kg, SECA 700, Seca GmbH, Hamburg, Germany).

### 2.3. Study of Osteoporotic Fractures (SOF) Index

As defined by the Study of Osteoporotic Fractures (SOF) Index, frailty was identified by the presence of two or more of the following three components: a weight loss of ≥5% during the preceding year (regardless of any intention to lose weight), the inability to rise from a chair five times without using the arms, and an answer of “no” to the question “Do you feel full of energy?”. Patients with none of the above impairments were considered robust, and those with one disability were considered to be in a pre-frailty/frailty status [20,21]. The SOF Index, a validated tool to assess frailty with an excellent correlation with outcomes in several settings [22,23], was administered during hospitalization after the stabilization of the clinical conditions. All the recruited subjects (100%) answered the SOF questionnaire and were included in the study.

### 2.4. Statistical Analysis

Continuous variables are expressed as mean ± standard deviation. Data were analyzed using one-way ANOVA, followed by Bonferroni’s post hoc analysis or unpaired t-test, as appropriate. The normal data distribution was evaluated using the Kolmogorov–Smirnov test. Continuous variables that were not normally distributed were natural log-transformed. Categorical variables are expressed as proportions and compared by the use of the chi^2^ test. To identify the predictors of the pre-frail/frail status, a logistic regression analysis was performed. A variance inflation factor (VIF) was used to identify the degree of multicollinearity. Because of an observational study design, and the absence in the literature of similar articles, we could not perform a sample size calculation. However, to check the statistical power of the results, we used a post hoc power analysis. The estimated power was calculated by considering the FEV1 means in the groups stratified by frailty (Appendix A). All data were collected in an Excel database and analyzed using STATA v 16 (StataCorp LLC, College Station, TX, USA). Significance was set at a level of *p* < 0.05.

## 3. Results

The study population consisted of 139 ambulatory CF patients (mean age 32.89 ± 10.94 years old; range 18–67), with 46% being female subjects, referred to the Cystic Fibrosis Centre for Adults of the University Federico II of Naples. The main characteristics of the total study population are reported in the Appendix A. All the data of the recruited subjects were included in the analyses.

Based on the SOF Index, the participants were identified as either robust (*n* = 84) or pre-frail/frail (*n* = 55). Most of the study population was robust (60.4%). Pre-frail/frail participants were more frequently females (*p* = 0.020) and had a lower BMI (*p* = 0.001), with worse respiratory function as shown by the spirometry parameters, and a higher number of pulmonary exacerbation/year, cycles of antibiotic therapy, and hospitalizations (all *p* < 0.001, Table 1) with respect to the robust patients. Moreover, the pre-frail/frail subjects used more drugs and were affected by more CF-related diseases (all *p* < 0.001). No significant differences were found between the groups regarding age, the main genotypes, the number of non-CF-related diseases, and smokers (Table 1).

*Staphylococcus aureus* was the most frequent infection affecting CF patients, without differences between the groups. Conversely, the pre-frail/frail subjects suffered from *Pseudomonas aeruginosa* (*p* < 0.001) and *Achromobacter xylosoxidans* (*p* = 0.022) infections significantly more frequently than robust subjects (Figure 1).

Table 2 and Table 3 show the main treatments and the main CF-related diseases, respectively. Aerosol therapy was the main treatment administered to both robust and pre-frail/frail subjects (Table 2). However, the pre-frail/frail group was the most frequent user of all the treatments except for thyroid hormone replacement therapy and corticosteroids.

All subjects in therapy with CFTR modulators (*n* = 29, 20.86%, Table 2) take 1 molecule of a combination of ivacaftor 75 mg/Tezcaftor 50 mg/elexacaftor 100 mg in the morning, and 1 pill of ivacaftor 150 mg in the evening, 12 h after the previous administration, every day.

Bronchiectasis and pansinusitis were the most frequent CF-related diseases in all the groups, with a higher rate in the frail group in respect to both pre-frail and robust subjects (*p* < 0.001, Table 3).

Then, by using the genotype classification, no differences were found among the groups. The pre-frail/frail subjects more frequently showed a CF-causing/CF-causing profile compared to the robust patients (Figure 2).

Moreover, to identify the best predictive factors for pre-frail/frail status, a logistic multivariate regression analysis was performed by using the frailty status as the dependent variable (Table 4). The independent variables introduced in the model were represented by the variables that were significant in the univariate analysis. VIF was used to identify the degree of multicollinearity and to exclude the multicollinear variables. Then, the final model included the following: gender, BMI, FEV1, Number of pulmonary exacerbations, Cycles of IV antibiotics therapy, Number of hospitalizations per year, BADL lost, IADL lost, Number of drugs, Number of CF-related diseases, and Number of non-CF-related diseases. The fitted regression model was 1.134 − 0.010 × (BMI) − 0.092 × (Women) − 0.221 × (FEV1) + 0.075 × (Number of pulmonary exacerbations) − 0.003 × (Cycles of IV antibiotics therapy) − 0.122 × (Number of hospitalizations per year) − 0.088 × (BADL) + 0.073 × (IADL) − 0.010 × (Number of drugs) + 0.030 × (Number of CF-related diseases) + 0.021 × Number of no CF-related diseases. The overall regression was statistically significant (r^2^ = 0.534, F (11, 126), *p* < 0.001). The best predictor of pre-frail/frail status was a low FEV1 value (β = −0.221, 95%CI −0.297 −0.145, *p* < 0.001), followed by the number of pulmonary exacerbations (β = 0.075, 95%CI 0.024 0.125, *p* = 0.004) (Table 4). This finding confirms that pre-frail/frail status is predicted by a poorer lung function measured by FEV1, suggesting CF as a geriatric syndrome.

## 4. Discussion

In this study, patients with CF were assessed for frailty, a geriatric syndrome that shares some features with this condition (sarcopenia, frequent hospitalization, etc.). We found that a pre-frail/frail phenotype was more frequently associated with a CF-causing/CF-causing profile, suggesting that CF could be considered an early representation of geriatric frailty syndrome. Indeed, in our study, CF more frequently affected females, and was characterized by a lower BMI and a greater number of drugs and hospitalization. Several studies have demonstrated that frailty syndrome affects women more frequently than men [24,25] by having a negative impact on the survival rate [25].

Previously, only few studies have investigated the relationship between frailty syndrome and CF, and often, the evaluation was performed on small samples and after lung transplantation (LT). In all cases, no evaluation of the association between the CF genotype classification and the frailty phenotype was presented.

Recently, Koutsokera et al. [26], by using a deficit accumulation approach, developed a CF-specific frailty index to allow for risk stratification for adverse waitlist and post-LT outcomes. To achieve this aim, they recruited 282 adult CF patients waitlisted for LT in two centers. Comorbidities, treatment, laboratory results, and social support at waitlisting were utilized to develop a lung disease severity index, a frailty index, and a lifestyle/social vulnerability index. A higher frailty index was significantly associated with a worsening waitlist status, post-LT mortality, and graft failure. The authors underlined that the frailty index was the first factor described to allow for graded risk stratification for CF patients waitlisted for LT [26].

Perez et al. [27] also supported frailty as a novel determinant of disability and quality of life in patients with CF undergoing lung transplantation. The authors demonstrated that, among 23 participants with CF, lung transplantation results in marked improvements in disability and quality of life. They found that four out of five patients who were frail before lung transplant were no longer frail six months after lung transplant. Further, improvements in frailty as well as in lung function were independently associated with improvements in disability and quality of life. Finally, they found that changes in BMI were not associated with changes in disability or quality of life, suggesting that the impact of frailty after lung transplantation cannot be explained by improvements in BMI [27].

In our study, we did not consider the changes over time of frailty, but we assessed this condition in non-transplanted CF subjects. Here, the pre-frail/frail CF patients showed higher disability, as shown by the significant differences in ADL and IADL scores but also by the doubled number of CF-related diseases in respect to the robust CF subjects.

Ferguson et al. [28] found that the frail patients suffered from more CF-related comorbidities, required more frequent courses of IV antibiotics, were older than the robust CF population, and exhibited significantly poorer lung function (FEV1), a well-recognized marker of CF severity [29]. Therefore, the authors suggested the use of frailty assessment criteria as a screening tool to identify CF patients who are most likely to suffer from future poor health [29]. By a logistic multivariate analysis, we also found that the best predictor of pre-frail/frail status was represented by low FEV1 values. Although there were some similarities with our results, the Ferguson study was strongly limited by the small sample size of the study population (18 patients) and the small number of considered variables without a multivariate analysis to assess the predictors of the condition.

In our study, we also analyzed the main causes of infections stratified by frailty status. We found that Staphylococcus aureus showed the highest prevalence without any difference between the groups. This datum is in accordance with the Patient Registry Annual Data Report 2021 by the Cystic Fibrosis Foundation [30] that showed the most prevalent microorganism in CF cases is Staphylococcus aureus (68%), while the percentage of individuals with a positive culture for *Pseudomonas aeruginosa* has continued to decline over time [30]. In our study, we observed significant differences in the prevalence of *Pseudomonas aeruginosa* and that it was more prevalent in pre-frail/frail than robust CF patients. This finding might also explain the differences obtained in lung functional parameters between the groups. A possible explanation of this datum might be related to the common substrate that characterizes the condition of fragility and the mechanisms put in place by *Pseudomonas aeruginosa* to ensure its survival. In fact, *P. aeruginosa* expresses multiple virulence factors during infection that enable the evasion of the host response. Some of these factors directly damage host tissue and subvert immune cell functions [31]. In turn, recent studies have provided a large body of evidence suggestive of a heightened inflammatory state in frail older individuals as marked by further increases of molecular and cellular inflammatory markers compared to those observed in robust older adult controls, suggesting multi-faceted dysregulation in both the innate and adaptive immune system in frail older adults [32]. Although the consequences of this chronic activation are unclear, it is conceivable that increased inflammation contributes to frailty through its detrimental effects on the physiology of organ systems [33]. 

The inflammatory response in CF lung disease has been recognized to have a central role in the pathological feature. An altered airway surface environment is pivotal in establishing and maintaining the inflammatory response in CF subjects [34]. However, several researchers have also identified a large number of underlying abnormalities in the CF airway inflammatory response that begin early in life [34].

Moreover, by using the genotyping classification, while we found a higher prevalence of the CF-causing/CF-causing genotype in the pre-frail/frail group, we did not find any differences between the pre-frail/frail and robust phenotype.

Finally, a multivariate analysis established FEV1 and the pulmonary exacerbations, the main CF characteristics, as the best predictors of the frailty syndrome, confirming the tight association between these two conditions. 

In agreement with our findings but under different conditions, in a meta-analysis evaluating the impact of frailty on health outcomes in people with Chronic Obstructive Pulmonary Disease (COPD), Wang et al. found that frail compared to non-frail subjects had a lower predicted Forced Expiratory Volume in the first second [35]. Hanlon et al. confirmed these results, underlining that the frailty phenotype (pre-frail/frail vs. robust), but not the frailty index, was associated with a lower FEV1 value [36]. 

### Limitations

The main limitation was the observational design of the study which did not allow us to estimate how changes in the frail condition over time could influence the main outcomes (mortality, hospitalization, etc.).

Another limitation is the lack of correlation between genotyping classification and the frailty phenotype. However, there is still no gold standard classification for CF because of the extreme heterogenicity in the clinical and genetic manifestation of this condition. Similarly, there is still no gold standard to determine a frail condition. However, the SOF Index tool used to define frailty in this study is a well-recognized instrument that has shown good correlation with the main health outcomes in several different settings [22,23].

## 5. Conclusions

In our study, we demonstrated that CF patients show similarities to pre-frail/frail older subjects, suggesting that CF might be considered an early expression of this geriatric syndrome. This finding is interesting because it could help to better define the possible progression of CF, but overall, it also suggests the usefulness of employing some of the tools used in the management and therapy of frailty subjects to identify the more severe CF subjects. Surely, these results need to be confirmed by other more extensive trials, also considering the impact of the new CF transmembrane regulator modulators that have demonstrated [37,38] their ability to substantially modify the progression and the management of the disease.

## Figures and Tables

**Figure 1 jcm-13-00585-f001:**
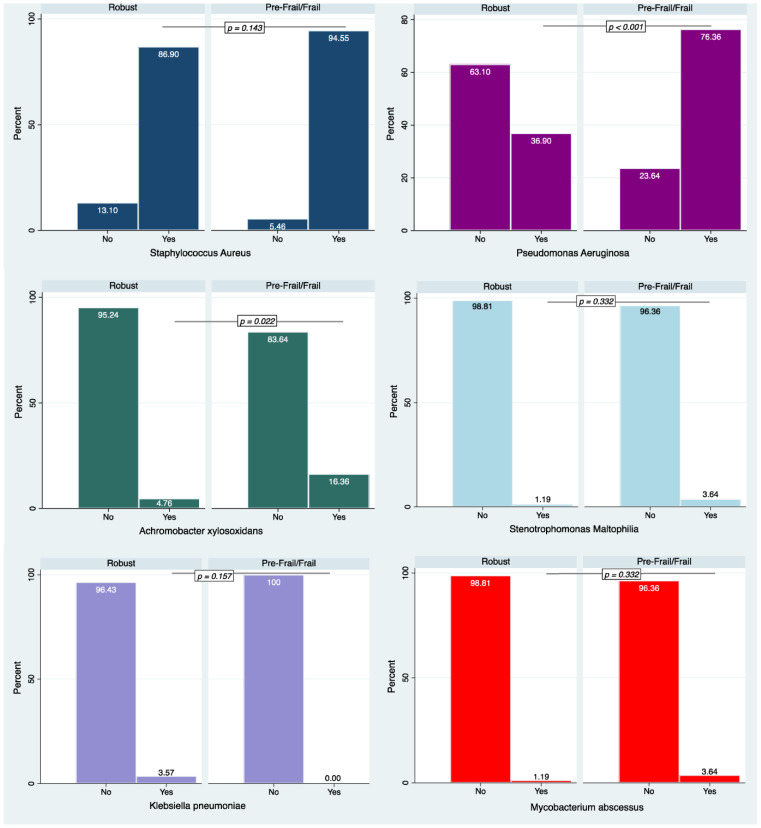
Prevalence of the main causes of infections in the study population stratified by frailty status.

**Figure 2 jcm-13-00585-f002:**
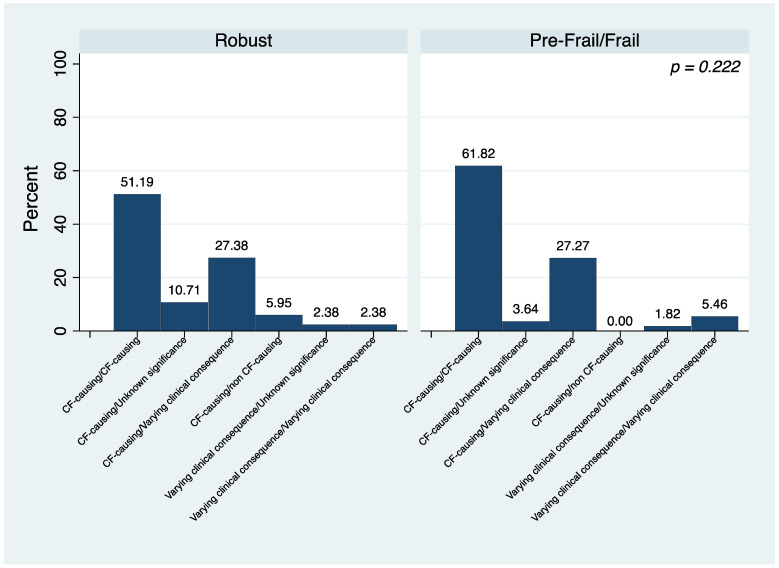
Differences in genotyping classification of the study population by frailty status.

**Table 1 jcm-13-00585-t001:** Main characteristics of the study population stratified by frailty status.

Variables	Robust (*n* = 84)	Pre-Frail/Frail (*n* = 55)	*p*
Age, years, mean ± SD	32.38 ± 11.48	33.67 ± 10.12	0.498
Gender, M/W, *n* (%)	52/32 (61.90/38.10)	23/32 (41.82/58.18)	**0.020**
BMI, kg/m^2^, mean ± SD	24.82 ± 3.76	22.57 ± 4.03	**0.001**
Smokers, *n* (%)	6 (7.14)	1 (1.82)	0.160
FEV_1_, L, mean ± SD	3.28 ± 0.98	1.55 ± 0.75	**<0.001**
FEV_1_, %, mean ± SD	84.93 ± 23.95	46.98 ± 22.65	**<0.001**
FVC, L, mean ± SD	4.26 ± 1.07	2.43 ± 0.86	**<0.001**
FVC, %, mean ± SD	94.95 ± 16.09	61.71 ± 20.45	**<0.001**
MMEF, %, mean ± SD	74.37 ± 32.43	29.33 ± 27.65	**<0.001**
**Main Genotypes**			0.116
DELTAF508/DELTAF508, *n* (%)	10 (12.05)	17 (30.91)	
DELTAF508/5T-12TG, *n* (%)	10 (12.05)	1 (1.82)	
DELTAF508/N1303K, *n* (%)	1 (0.72)	4 (2.90)	
N° pulmonary exacerbations per year, mean ± SD	0.58 ± 1.06	2.56 ± 2.51	**<0.001**
Cycles of IV antibiotic therapy, mean ± SD	0.08 ± 0.28	0.58 ± 1.32	**<0.001**
Cycles of oral antibiotic therapy, mean ± SD	0.50 ± 0.91	1.93 ± 1.87	**<0.001**
N° of hospitalizations per year, mean ± SD	0.11 ± 0.35	0.47 ± 0.88	**<0.001**
BADL lost, mean ± SD	0.00 ± 0.00	0.24 ± 0.77	**0.006**
IADL lost, mean ± SD	0.00 ± 0.00	0.65 ± 1.82	**0.001**
N° of drugs, mean ± SD	4.49 ± 3.80	8.60 ± 3.10	**<0.001**
N° of CF-related diseases, mean ± SD	2.65 ± 2.57	5.67 ± 2.64	**<0.001**
N° of non-CF-related diseases, mean ± SD	0.35 ± 0.67	0.42 ± 0.74	0.547
N° of total diseases, mean ± SD	3.00 ± 2.57	6.09 ± 2.80	**<0.001**

BMI, Body Mass Index; FEV_1_, Forced Expiratory Volume in the first second; FVC, Forced Vital Capacity; MMEF, Maximal Mid-Expiratory Flow; IV, Intravenous; BADL, Basic Activity of Daily Living; IADL, Instrumental Activity of Daily Living; CF, Cystic Fibrosis. In bold the statistically significant differences are reported.

**Table 2 jcm-13-00585-t002:** Main treatments of the study population stratified by frailty status.

Variables, *n* (%)	Total (*n* = 139)	Robust (*n* = 84)	Pre-Frail/Frail (*n* = 55)	*p*
Aerosol therapy	111 (79.86)	60(71.40)	51(94.44)	**0.001**
Bronchodilators	107 (76.98)	55 (65.48)	52 (94.55)	**<0.001**
Antibiotics via aerosol	90 (64.75)	41 (48.81)	49 (89.09)	**<0.001**
Nasal washes	88 (63.31)	44(52.38)	44(80.0)	**0.001**
Ursodeoxycholic acid	82 (58.99)	36 (42.86)	46 (83.64)	**<0.001**
Liposoluble vitamins	81 (58.27)	36 (42.86)	45 (81.82)	**<0.001**
Pancreatic enzymes	79 (56.83)	33(39.29)	46 (83.64)	**<0.001**
Azithromycin	53 (38.13)	19 (22.62)	34 (61.82)	**<0.001**
Gastroprotectors	44 (31.65)	15 (17.86)	29 (52.73)	**<0.001**
CFTR modulators	29 (20.86)	11 (13.10)	18 (32.73)	**0.005**
Insulin	28 (20.14)	11 (13.1)	17 (30.91)	**0.010**
Antimycotics	25 (17.99)	10 (11.90)	15 (27.27)	**0.021**
Oxygen therapy	11 (7.91)	0 (0.00)	11 (20.00)	**<0.001**
Corticosteroids	8 (5.76)	3(3.6)	5 (9.09)	0.172
Thyroid Hormonal replacement	8 (5.76)	3 (3.57)	5 (9.09)	0.172
Bisphosphonates	6 (4.32)	0 (0.00)	6 (10.91)	**0.002**

CFTR, Cystic Fibrosis Transmembrane-conductance Receptor. In bold the statistically significant differences are reported

**Table 3 jcm-13-00585-t003:** Main CF-related diseases of the study population divided by frailty status.

Variables, *n* (%)	Total (*n* = 139)	Robust (*n* = 84)	Pre-Frail/Frail (*n* = 55)	*p*
Bronchiectasis	96 (69.06)	46 (54.76)	50 (90.91)	<0.001
Pansinusitis	90 (64.75)	44 (52.38)	46 (83.64)	<0.001
Exocrine pancreatic insufficiency	76 (54.68)	33(39.29)	43 (78.18)	<0.001
CBAVD	47 (33.81)	26 (30.95)	21 (38.18)	0.378
CFRD	37 (26.62)	13 (15.48)	24 (43.64)	<0.001
Chronic cholestatic hepatopathy	35 (25.18)	13 (15.48)	22 (40.00)	0.001
Chronic respiratory failure	29 (20.86)	6 (7.14)	23 (41.82)	<0.001
Hemoptysis	28 (20.14)	8 (9.52)	20 (36.36)	<0.001
Arterial hypertension	18 (12.95)	12 (14.29)	6 (10.91)	0.562
ABPA	14 (10.07)	4 (4.76)	10 (18.18)	0.010
Osteoporosis	13 (9.35)	1 (1.19)	12 (21.82)	<0.001
Nasal polyps	12 (8.63)	10 (11.9)	2 (3.64)	0.090
DIOS	12 (8.63)	5 (5.95)	7 (12.73)	0.164
Hypothyroidism	12 (8.63)	4 (4.76)	8 (14.55)	0.045
Pneumothorax	8 (5.76)	1 (1.19)	7 (12.73)	0.004
Recurrent pancreatitis	6 (4.32)	4 (4.76)	2 (3.64)	0.750
Arthritis	5 (3.62)	2(2.38)	3 (5.56)	0.330
Dyslipidemia	5 (3.62)	4 (4.76)	1 (1.85)	0.372
Liver steatosis	4 (2.88)	2 (2.38)	2 (3.64)	0.665
Cirrhosis	4 (2.88)	1 (1.19)	3 (5.45)	0.141
Cholelithiasis	4 (2.88)	2 (2.38)	2 (3.64)	0.665
Salt wasting syndrome	3 (2.16)	1 (1.19)	2 (3.64)	0.332
Thyroiditis	3 (2.16)	3 (3.57)	0 (0.00)	0.157
Nephrolithiasis	2 (1.45)	1 (1.19)	1 (1.85)	0.751
Multinodular goiter	2 (1.44)	0 (0.00)	2 (3.64)	0.078
GERD	2 (1.44)	1 (1.19)	1 (1.82)	0.761
Celiachia	2 (1.44)	1 (1.19)	1 (1.82)	0.761
Chronic kidney failure	1 (0.72)	1 (1.19)	0 (0.00)	0.421
Depression	1 (0.72)	0 (0.00)	1 (1.85)	0.211
Behavioral disturbances	1 (0.72)	0 (0.00)	1 (1.85)	0.211
Allergy	1 (0.72)	1 (1.19)	0 (0.00)	0.421
Lupus erythematosus systemic	1 (0.72)	0 (0.00)	1 (1.85)	0.211
Urinary tract tumors	1 (0.73)	1 (1.20)	0 (0.00)	0.418
Breast cancer	1 (0.72)	1 (1.19)	0 (0.00)	0.421
Thyroid tumors	1 (0.72)	0 (0.00)	1 (1.85)	0.211

CBAVD, Congenital Bilateral Absence of the Vas Deferens; CFRD, Cystic fibrosis-related diabetes; ABPA, allergic bronchopulmonary aspergillosis; DIOS, distal intestinal obstruction syndrome; GERD, Gastro-Esophageal Reflux Disease.

**Table 4 jcm-13-00585-t004:** Logistic multivariate regression analysis.

Robust vs. Pre-Frail/Frail	β	95% Conf. Interval Low High	*p*
Gender			
Women	−0.092	−0.236 0.051	0.205
BMI, kg/m^2^	−0.010	−0.027 0.007	0.261
FEV_1_, L	−0.221	−0.297 −0.145	**<0.001**
Number of pulmonary exacerbations per year	0.075	0.024 0.125	**0.004**
Cycles of IV antibiotic therapy	−0.003	−0.149 0.142	0.966
Number of hospitalizations per year	−0.122	−0.326 0.082	0.239
BADL lost	−0.088	−0.293 0.117	0.399
IADL lost	0.073	−0.004 0.150	0.063
Number of drugs	−0.010	−0.040 0.019	0.483
Number of CF-related diseases	0.021	−0.018 0.060	0.287
Number of non-CF-related diseases	0.030	−0.062 0.121	0.525

BMI, Body Mass Index; FEV_1_, Forced Expiratory Volume in 1 s; BADL, Basic Activity of Daily Living; IADL, Instrumental Activity of Daily Living; CF-related, Cystic Fibrosis-related. The statistically significant values are reported in bold.

## Data Availability

The data presented in this study are available upon request from the corresponding author. The data are not publicly available due to privacy concerns.

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
