# Peer review of "Cystic Fibrosis in Adults: A Paradigm of Frailty Syndrome? An Observational Study"

_jcm, 2024, doi:10.3390/jcm13020585_

Round 1

Reviewer 1 Report

Comments and Suggestions for Authors

I have read with great interest the study by Lacotucci et al investigating the characteristics of adult Cystic Fibrosis (CF) patients and the association of frailty with the CF genotyping classification. Overall, the paper is well-written and concise and adds useful information to the everyday clinical practice.

The major findings of the study seem to be in accordance with clinical experience. Frailty as the expression of the underlying inflammation has multi-systemic effects. For example, patients often have a decline in the ability to cough and weak cough diminishes the ability of airway clearance. Personally, I find quite interesting the finding that in logistic regression the best predictor of the Pre-Frail/Frail status was the low FEV1 level. This is in agreement with similar findings in patients with other degenerative diseases of the lower respiratory system such as COPD. Indeed, COPD patients with frailty had lower predicted FEV1 values (Wang et al. 2023).

Moreover, the frailty phenotype (frail vs robust), but not the frailty index, was associated with lower FEV1.

Wang L, Zhang X, Liu X. Prevalence and clinical impact of frailty in COPD: a systematic review and meta-analysis. BMC Pulm Med. 2023 May 12;23(1):164. doi: 10.1186/s12890-023-02454-z. PMID: 37173728.

Hanlon P, Lewsey J, Quint JK, Jani BD, Nicholl BI, McAllister DA, Mair FS. Frailty in COPD: an analysis of prevalence and clinical impact using UK Biobank. BMJ Open Respir Res. 2022 Jul;9(1): e001314. doi: 10.1136/bmjresp-2022-001314. PMID: 35787523.

Author Response

Please find in the attachment, the point-by-point response.

Reviewer 2 Report

Comments and Suggestions for Authors

Dear authors, thank you for your article. I have some recommendations, which I hope would improve the article. 

Introduction section - you say that 3. Sweat chloride 48 value 40 to 59 mmol/L with no or 1 CF-causing mutation, but with family history and/or 49 ancillary testing and clinical presentation strongly suggestive of CF and quote an old guideline from 2008.

However, in this guideline it stated that Individuals with no or 1 CF-causing mutation and clinical findings suggestive of CFTR dysfunction (i.e., obstructive azoospermia, bronchiectasis, or acute, recurrent, or chronic pancreatitis) may be diagnosed with a CFTR-related disorder, depending on their clinical picture or family history, and are at risk for CF. Please, correct your introduction section, since CF is still considered an autosomal recessive disorder and two mutations in the CFTR gene are needed. The one mutation you are referring could be defined as CFTR-related disorder. 

Study population - more information about the types of the disease-causing mutations should be given.

Results - did some of the included subjects received other therapy, like ivacaftor? 

Discussion - I have no recommendations. 

I congratulate you on your nice figures.

All the best!

Author Response

(The authors gave the same response as above.)

Reviewer 3 Report

Comments and Suggestions for Authors

Dear authors

I would like to thank you for giving me the opportunity to review the manuscript entitled “Cystic Fibrosis in Adults: A Paradigm of Frailty Syndrome? An Observational Study”. The aim of this study was to assess the main clinical and anamnestic characteristics of adult Cystic Fibrosis (CF) patients and to evaluate the association of frailty with the CF genotyping classification. This manuscript is well-designed and -written and provides valuable information that can contribute to future direction to improve the safety and quality of life of people with CF. I have some comments that you can be considered:

- Please apply the full form of abbreviations in the first use

- In the introduction, more review of the literature is needed to justify the conduction of this study.

- How was the required number of samples determined?

- How did you approach the participants and how was the sampling done? It must be fully explained

- The way of sampling and completing the questionnaires should be explained in more detail.

- Please move the table related to logistic multivariate regression analysis to the main text rather than a supplementary file.

Author Response

(The authors gave the same response as above.)

Round 2

Reviewer 3 Report

Comments and Suggestions for Authors

Dear author 

Thank you for addressing my comments.